# HSPA5 Promotes the Proliferation, Metastasis and Regulates Ferroptosis of Bladder Cancer

**DOI:** 10.3390/ijms24065144

**Published:** 2023-03-07

**Authors:** Qinghua Wang, Shuai Ke, Zelin Liu, Haoren Shao, Mu He, Jia Guo

**Affiliations:** 1Department of Urology, Renmin Hospital of Wuhan University, Wuhan 430060, China; 2Britton Chance Center for Biomedical Photonics, Wuhan National Laboratory for Optoelectronics, Huazhong University of Science and Technology, Wuhan 430074, China

**Keywords:** HSPA5, bladder cancer, proliferation, metastasis, VEGFA/VEGFR2, ferroptosis

## Abstract

Heat shock protein family A (HSP70) member 5 (HSPA5) is aberrantly expressed in various tumors and closely associated with the progression and prognosis of cancer. Nevertheless, its role in bladder cancer (BCa) remains elusive. The results of our study demonstrated that HSPA5 was upregulated in BCa and correlated with patient prognosis. Cell lines with low expression level of HSPA5 were constructed to explore the role of this protein in BCa. HSPA5 knockdown promoted apoptosis and retarded the proliferation, migration and invasion of BCa cells by regulating the VEGFA/VEGFR2 signaling pathway. In addition, overexpression of VEGFA alleviated the negative effect of HSPA5 downregulation. Moreover, we found that HSPA5 could inhibit the process of ferroptosis through the P53/SLC7A11/GPX4 pathway. Hence, HSPA5 can facilitate the progression of BCa and may be used as a novel biomarker and latent therapeutic target in the clinic.

## 1. Introduction

Bladder cancer has become the most common malignancy of the urinary system based on the estimation by the American Cancer Society [1]. Overall, it has been projected that approximately 81,180 new BCa cases will be diagnosed and 17,100 patients will die because of BCa in the United States in 2022. Tobacco plays a vital role in the induction of BCa; in addition, occupational exposure to hydrocarbons, genetic predisposition and drinking water containing arsenic are all risk factors [2]. Most BCa cases are diagnosed as non-muscle-invasive bladder cancer (NMIBC), which has a better prognosis than muscle-invasive bladder cancer (MIBC) [3,4]. Although there have been great technological advances, there has been little progress in BCa treatment in recent years [5,6,7]. Hence, developing an effective and appropriate treatment method is extremely important for the prognosis of BCa patients.

Heat shock protein family A (HSP70) member 5 (HSPA5), also called 78 kDa glucose-regulated protein (GRP78), is located at 9q33.3 and has 8 exons [8]. HSPA5 is involved in various human diseases, such as infection and tumorigenesis. Studies have shown that HSPA5 is associated with COVID-19 as a potential SARS-CoV-2 receptor in host cells [9]. In addition, aberrant expression of HSPA5 promotes the progression of breast cancer, pancreatic cancer, and other cancers [10,11,12,13,14]. Furthermore, HSPA5 can be regulated by noncoding RNAs, such as miR-30 and miR-1199-5p [15,16]. Interestingly, HSPA5 is also involved in ferroptosis and plays a vital role in ferroptosis resistance [17]. Nevertheless, the function of HSPA5 in BCa remains elusive and needs to be explored.

Vascular endothelial growth factor A (VEGFA) belongs to the VEGF family, which also includes VEGFB, VEGFC, VEGFD, VEGFE and placenta growth factor (PGF) [18]. VEGFA plays a critical role in cancer by promoting the proliferation, angiogenesis and metastasis of cancer cells [19,20]. Aberrant expression of VEGFA can activate many signaling pathways by targeting VEGFR, such as PI3K/AKT and ERKs [21,22].

Ferroptosis, a unique type of programmed cell death, was described by Dixon in 2012 and is associated with a variety of human diseases including cancer [23,24]. Aberrant lipid peroxidation and iron dependency are commonly considered the major characteristics of ferroptosis [25]. Morphologically, ferroptotic cells display changes in mitochondrial ultrastructure, such as a smaller volume, increased mitochondrial membrane density and diminished or absent mitochondrial cristae [25]. In addition, increasing in lipid ROS and Fe^2+^, dysfunction of the cystine/glutamate antiporter (xCT system) and glutathione peroxidase 4 (GPX4) have been observed during the process of ferroptosis [26]. These features can be used to distinguish ferroptosis from other forms of death, such as apoptosis, autophagy, pyroptosis, and necrosis [27]. Additionally, many studies have reported that various factors can trigger or inhibit ferroptosis, including proteins, noncoding RNAs and compounds, which may suggest ferroptosis as a potential therapeutic target for clinical treatment [28,29].

Here, the results of our study demonstrate that HSPA5 is elevated in BCa tissues and significantly associated with tumor progression and poor prognosis in BCa patients. In addition, the biological function of HSPA5 in the progression of BCa was explored and knockdown of HSPA5 significantly inhibits the proliferation and metastasis of BCa cells via blocking the VEGFA/VEGFR2 pathway. Furthermore, knockdown of HSPA5 triggers the progression of ferroptosis and sensitizes BCa cells to ferroptosis inducers through the P53/SLC7A11/GPX4 pathway.

## 2. Results

### 2.1. The Expression Levels of HSP70s in Bladder Cancer

To explore the expression levels of HSP70s in BCa, we analyzed the levels of HSPA1A, HSPA1B, HSPA1L, HSPA2, HSPA5, HSPA6, and HSPA8 in BCa from The Cancer Genome Atlas (TCGA) database. As the results show, the difference in the expression of HSPA2 and HSPA5 were the most obvious from both unpaired samples and paired samples of BCa patients (Figure 1A,B). To further validate our analysis, StarBase and UALCAN databases were used, and these results were consistent with those above (Figure 1C,D). Next, we also explored HSPA2 and HSPA5 expression based on nodal metastasis status and individual cancer stage (Figure 1E,F). Above all, HSPA2 and HSPA5 expression levels were found to be significantly downregulated or upregulated in cancer tissues compared with normal tissues.

### 2.2. The Relationship between HSPA2 and HSPA5 Expression and Patient Prognosis

To investigate the role of HSPA2 and HSPA5 in the prognosis of human BCa, we found that patients with high level of HSPA5 had poor prognosis from the TCGA database heatmap (Figure 2A). In Figure 2B,C, the overall survival analysis from GEPIA and the Kaplan-Meier plotter demonstrated that patients with a high level of HSPA5 had a lower overall survival rate, while the level of HSPA2 had no effect on overall survival. Furthermore, HSPA5 expression was negatively correlated with the disease-free survival rate of BCa patients, which indicated patients with higher HSPA5 expression had poorer prognosis (Figure 2D–F). Based on the above results, we concluded that HSPA5 was upregulated in BCa samples, as well as negatively correlated with poor prognosis of BCa patients.

### 2.3. The Expression of HSPA5 in Clinical Specimens and Cell Lines

IHC was performed to further explore the expression of HSPA5 in clinical BCa patients. The results of IHC indicated that HSPA5 expression level in BCa tissues was significantly higher than that in adjacent tissues (Figure 3A). Western blotting also demonstrated that HSPA5 was upregulated in BCa samples compared with adjacent samples (Figure 3B,C). In addition, we found that the levels of HSPA5 in T24, EJ and 5637 cell lines were higher than that in SV-HUC-A cell (Figure 3D,E). Because the expression difference in T24 and 5637 was more obvious, the two cell lines were used to construct cell lines with low expression level of HSPA5. The efficiency of HSPA5 knockdown was verified by qRT-PCR and Western blotting; both the mRNA and protein expression of HSPA5 were significantly downregulated in the sh-HSPA5 groups compared with the control groups (Figure 3F,G).

### 2.4. Downregulating HSPA5 Inhibits BCa Cell Growth and Metastasis In Vitro

The biological effects of HSPA5 on tumor cells and the underlying mechanisms have not been clarified. To investigate if HSPA5 is involved in the process of BCa cells, we first probed the role of HSPA5 in BCa cell proliferation and a colony formation assay was performed. As shown in Figure 4A,B, the number of cell colony formation was dramatically reduced in the sh-HSPA5 groups compared with the control groups. Next, the CCK-8 assay was utilized to detect cell viability, and the results were consistent with the colony formation data (Figure 4C). Furthermore, the decreased cell proliferation ability was verified by an EdU assay (Figure 4D). In addition, wound-healing assay (Figure 4E,F) and migration assay (Figure 4G,H) demonstrated that low HSPA5 expression inhibited BCa cells migration. Moreover, the number of invading cells decreased in the sh-HSPA5 groups (Figure 4I,J). Based on the above observations, downregulating HSPA5 could inhibit the proliferation and metastasis of BCa cells.

### 2.5. Effects of HSPA5 on the Cell Cycle and Apoptosis in BCa Cells

To further investigate the role of HSPA5 in the proliferation of BCa cells, we analysed the effect of HSPA5 downregulation on the cell cycle in BCa cells. The results demonstrated that downregulation of HSPA5 increased the percentage of T24 and 5637 cells in S phase (Figure 5A,B). Moreover, a low expression level of HSPA5 elevated the percent of apoptotic cells (Figure 5C,D). Western blotting analysis proved these results (Figure 5E,F). Cell cycle-related proteins, such as CDK2 and cyclin A, were downregulated when HSPA5 was knocked down. The expression of Bax and cleaved caspase-3 was significantly elevated, while Bcl-2 was downregulated in the sh-HSPA5 cell groups. These results demonstrated that knockdown of HSPA5 inhibited cell proliferation by inducing S phase arrest and promoting apoptosis.

### 2.6. Inhibiting HSPA5 Suppresses BCa Progression by Downregulating the VEGFA/VEGFR2 Pathway

Next, we set out to explore the molecular mechanism by which HSPA5 induces cell proliferation and metastasis, and the signaling pathway enrichment analysis was carried out. We found that HSPA5 may promote BCa progression via the VEGFA/VEGFR2 signaling pathway (Figure 6A). To verify this hypothesis, qRT-PCR and Western blotting were performed to measure the expression of VEGFA when HSPA5 was knocked down. Both the mRNA and protein levels of VEGFA were decreased in the sh-HSPA5 groups compared with the control (Figure 6B,C). Next, IHC was used and the result showed that VEGFA was upregulation in BCa samples (Appendix A). It has been reported that phosphorylated VEGFR2 also elevates the level of PI3K/AKT phosphorylation. As the results show, downregulation of HSPA5 reduced the levels of *p*-VEGFR2, *p*-PI3K, and *p*-AKT (Figure 6E,F). Next, we investigated whether HSPA5 affects the process of EMT. The expression of N-cadherin, vimentin, and Snail was significantly decreased; in contrast, E-cadherin was upregulated (Figure 6G,H). In addition, the immunofluorescence results confirmed the above conclusions (Figure 6I). In the next step, a cell line with overexpression VEGFA was constructed (Figure 7A,B). In these cells with the levels of *p*-PI3K and *p*-AKT decreased, this inhibitory effect was reversed after upregulation of VEGFA (Figure 7A,B). In the colony formation assay, the inhibitory effect caused by HSPA5 downregulation was attenuated when VEGFA was overexpressed (Figure 7E,F). In addition, the EdU results were consistent with the CCK-8 results (Figure 7G). Wound healing and migration assays also demonstrated that upregulation of VEGFA rescued the migration ability of BCa cells (Figure 7H–K). The invasion assay demonstrated that the number of invading cells was elevated when VEGFA was upregulated in the sh-HSPA5 cell lines (Figure 7L,M). In conclusion, the above results revealed that the upregulation of VEGFA attenuates the inhibitory effect of HSPA5 knockdown in BCa cells.

### 2.7. HSPA5 Inhibition Enhances BCa Cell Ferroptosis

Studies have demonstrated that HSPA5 is involved in ferroptosis in various cancers, so we sought to determine whether HSPA5 plays a vital role in BCa. First, RSL3 was found to inhibit the activity of bladder cancer cells in a concentration-dependent manner; in addition, this inhibitory effect could be blocked by a ferroptosis inhibitor but not an apoptosis inhibitor or necroptosis inhibitor (Figure 8A,B). Moreover, we found that HSPA5 expression was elevated after treatment with RSL3 (Figure 8C,D). In addition, HSPA5 downregulation increased BCa cell sensitivity to RSL3 and the growth inhibition after treatment with RSL3 could be reversed by Ferrostatin-1 (Appendix A). To explore the exact mechanism of by which HSPA5 mediates ferroptosis, the STRING database was utilized to identify the downstream targets of HSPA5. We found that HSPA5 could interact with P53, which can negatively regulate the SLC7A11/GPX4 signaling pathway (Figure 8E). Knockdown of HSPA5 increased the expression of P53, and the CO-IP results demonstrated that there was an interaction between HSPA5 and P53 (Figure 8F–I). Moreover, SLC7A11 and GPX4 expression was remarkably downregulated according to the extent of P53 elevation (Figure 8G,H). Morphologically, the cell mitochondria in the sh-HSPA5 group were smaller in volume, and their mitochondrial membrane density was increased (Figure 8J). Compared with the control groups, the level of ROS in the sh-HSPA5 cells was significantly higher. The MDA level also increased, while GSH was downregulated (Figure 8K,L). All results revealed that HSPA5 inhibition enhances BCa cell ferroptosis via the P53/SLC7A11/GPX4 pathway.

### 2.8. HSPA5 Knockdown Inhibits Xenograft Growth In Vivo

Finally, to explore the effect of HSPA5 on BCa cell growth and ferroptosis in vivo, T24 cells with and without HSPA5 knockdown were implanted under the skins of nude mice. The tumour volumes and weight of the mice were measured. As shown in Figure 9A,B, we found that the tumour volumes and weight in the sh-HSPA5 groups were smaller than that in the control group. Next IHC was performed, and the data demonstrated that BCa cell proliferation was suppressed because of the downregulation of HSPA5 (Figure 9C). Moreover, the IHC results indicated that Ki-67 expression was low in the sh-HSPA5 groups, which demonstrated that downregulation of HSPA5 inhibited tumour growth in vivo (Figure 9C). In addition, the expression of VEGFA, vimentin, and the ferroptosis-related protein SLC7A11 was decreased; in contrast, P53 was upregulated (Figure 9C). These results revealed that HSPA5 knockdown inhibited BCa cell growth and triggered ferroptosis (Figure 10).

## 3. Discussion

BCa has the highest rate of incidence among cancers of the urinary system, and it is estimated that there will be 81,180 new BCa cases and 17,100 deaths in the US in 2022 due to BCa [1]. Tobacco plays a very large role in the occurrence of BCa, and other factors include occupational exposure to hydrocarbons and genetic predisposition. In clinical practice, BCa can be classified as either NMIBC or MIBC [6]. Approximately 80% of BCa patients are diagnosed with NMIBC, which has shown a better life expectancy; the 5-year survival rate has not markedly changed despite advancements in diagnosis and care [30,31,32]. Therefore, finding new treatments for bladder cancer is urgent and necessary.

HSPA5 is also known as GRP78 or BIP, and it localizes to the lumen of the endoplasmic reticulum (ER). HSPA5 is involved in protein folding and assembly as a molecular chaperone of HSP70 in the ER and regulates ER homeostasis [8]. HSPA5 participates in ER stress by interacting with S100 calcium-binding protein 16 and activating IRE1α [33]. Additionally, HSPA5 also plays a crucial role in various cancers. In head and neck cancer, MUL1 induces the ubiquitination of HSPA5 at lysine 446, which triggers cell apoptosis [14]. HSPA5 is regulated by CD5L and promotes the proliferation of liver cancer cells, playing an antiapoptotic role [34]. In our present research, we revealed that HSPA5 is upregulated in BCa and that patients with high levels of HSPA5 expression tend to have a poor prognosis. To explore the precise mechanism of HSPA5 in BCa, the HSPA5 knockdown cell lines were constructed. The results demonstrated that the proliferation of BCa cells was blocked in the sh-HSPA5 groups compared with the control groups. In addition, migration and invasion were also retarded after downregulation of HSPA5.

The signaling pathway enrichment results indicated that HSPA5 may promote BCa progression by regulating the VEGFA/VEGFR2 signaling pathway. VEGFA is a member of the VEGF family and is crucial for both physiological and pathological angiogenesis [35]. Aberrant expression of VEGFA can induce the proliferation and migration of vascular endothelial cells. Furthermore, VEGFA is involved in the development of various tumors [36,37,38]. qRT-PCR and Western blotting were performed to verify whether HSPA5 could affect the expression of VEGFA. As the results show, the level of VEGFA mRNA was significantly downregulated in the sh-HSPA5 BCa cell lines, and these results were confirmed by the Western blotting data. Many studies have shown that VEGFA can interact with VEGFR2. Therefore, the level of p-VEGFR2 was measured. The results indicated that p-VEGFR2 was downregulated. Furthermore, the expression of p-PI3K and p-AKT was also decreased in the sh-HSPA5 groups compared to the control groups. Moreover, we found that upregulation of VEGFA could reverse the negative effect of HSPA5 knockdown, which further validated the above results.

Ferroptosis is a newly discovered kind of programmed cell death that is different from apoptosis, autophagy, pyroptosis and necrosis. Lipid peroxidation and aberrant iron accumulation are the major characteristics of ferroptosis in cells. Additionally, in terms of their morphology, the mitochondria has a smaller volume, increased membrane density and diminished or absent cristae in cells undergoing ferroptosis. In addition, many studies have reported that various factors can trigger or inhibit ferroptosis, which may suggest ferroptosis as a potential therapeutic target for clinical treatment. Studies have shown that HSPA5 is also associated with the progression of ferroptosis [17,39]. In BCa cells, we found that the expression of HSPA5 was elevated after treatment with RSL3, a ferroptosis inducer. Tumor suppressor P53 plays a vital role in tumor suppression and is also involved in many biological and pathological processes, such as apoptosis. In addition, there are also many articles showed that P53 can enhance ferroptosis by inhibiting the expression of SLC7A11 [40]. We found HSPA5 may regulate the level of P53 by the analysis of String Database. Knockdown of HSPA5 increased the expression of P53, and the CO-IP results demonstrated that there was an interaction between HSPA5 and P53. Moreover, SLC7A11 and GPX4 expression was remarkably downregulated according to the extent of P53 elevation. Additionally, the level of ROS in the sh-HSPA5 groups was higher than that in the control groups, while GSH was downregulated. In conclusion, HSPA5 inhibition increased BCa cell sensitivity to ferroptosis via the P53/SLC7A11/GPX4 pathway.

## 4. Materials and Methods

### 4.1. Patient Samples

All BCa tissues and adjacent tissues were obtained from the Renmin Hospital of Wuhan University (Wuhan, China) between February 2020 and February 2022, and all specimens were collected from patients without a prior history of BCa or adjuvant therapy. All patients signed the patient informed consent form, and all patients’ information was shown in Appendix A. In addition, all samples were fixed with 4% paraformaldehyde or placed in liquid nitrogen after surgery. Ethical approval was granted by the Ethics Committee of Renmin Hospital of Wuhan University.

### 4.2. Cell Culture and Treatment

BCa cell lines (T24, EJ, and 5637) and human normal bladder epithelium cell SV-HUC-1 were purchased from the Cell Bank of the Chinese Academy of Sciences (Shanghai, China). All BCa cells were cultivated in RPMI-1640 medium (HyClone, Logan, UT, USA) with 10% foetal bovine serum (FBS) (Gibco, Waltham, MA, USA), and SV-HUC-1 cells were maintained in F-12K medium (HyClone, USA) with 10% FBS. All cells were incubated in a humidified chamber containing 5% CO_2_ at 37 °C and medium was replaced every 2 d. RSL3 was purchased from MCE (Shanghai, China).

### 4.3. Cell Transfection Assay

To generate cells with HSPA5 knockdown, lentiviruses containing small hairpin RNA-HSPA5 (sh-HSPA5) and sh-Control were constructed (Viraltherapy, China). When the cell density was 60%, sh-HSPA5 or sh-Control was transfected into BCa cells by using Lipofectamine 2000 (Invitrogen, Carlsbad, CA, USA). The efficiency of transfection was verified by using qRT–PCR and Western blot analyses. The VEGFA overexpression model was constructed as described above.

### 4.4. Quantitative Real-Time PCR

TRI Reagent (Absin, China) was utilized for extracting cell RNA. cDNA was synthesized using ABScript II Reverse Transcriptase (ABclonal, Wuhan, China) following the manufacturer’s instructions with Veriti PCR instruction. 2× Universal SYBR Green Fast qPCR Mix (ABclonal, China) was used for qPCR with Lightcycler 4800II. Fold enrichment was calculated with the 2^−ΔΔCt^ method relative to the expression of GAPDH. All experiments were conducted in triplicate and repeated three times. All primers (Sangon Biotech, Shanghai, China) used are listed in the following Table 1:

### 4.5. Western Blotting

Cells were digested and centrifuged, and the cell pellet was left behind after removal of the supernatant. All samples were lysed on ice in RIPA lysis buffer (Servicebio, Wuhan, China) with phosphatase inhibitor and protease inhibitor. After 30 min, the lysates were centrifuged, and the supernatant was then collected. The remaining supernatant was added to 25% 5 × SDS-PAGE Sample Loading Buffer (ABclonal, China) and then boiled at 100 °C for 15 min. Equal amounts of protein were added to 8% or 12% SDS–PAGE gels and then transferred to the NC membrane (PALL, Port Washington, NY, USA), which was blocked with Protein Free Rapid Blocking Buffer (5×) (Epizyme, Shanghai, China) for 15 min. Next, primary antibodies and secondary antibodies were added individually for incubation. Bands were detected by chemiluminescence, and results were quantified with ImageJ-win64 software. All antibodies used are listed in the following Table 2:

### 4.6. Immunoprecipitation (IP)

The immunoprecipitation kit was purchased from Beyotime (Shanghai, China). Cells were lysed in RIPA lysis buffer on ice with protease inhibitor for 30 min, and the lysate was centrifuged and the suspension was conserved. IgG (Proteintech, Wuhan, China) or primary antibody was added into the suspension at 4 °C overnight, and the Protein A+G beads were added to the mixture for incubation. The supernatant was removed after centrifugation, all the beads were washed three times with the inhibitor-containing lysate. SDS-PAGE Sample Loading Buffer (1×) was then added to the samples and heated at 100 °C for 5 min. Finally, the supernatant was taken for Western blotting after separation.

### 4.7. CCK-8 Assay

Cells were seeded in 96-well plates (Corning, Corning, NY, USA) at approximately 3 × 10^3^ cells/well and cultured in humidified chamber. PBS was added around the 96-well plate to prevent medium volatilization. At the appointed time, 100 µL of culture medium with 10% CCK-8 was added to each well (Biosharp, Beijing, China). After incubation at 37 °C for 2 h, the OD value was measured. Each sample was repeated three times.

### 4.8. Colony Formation Assay

Cells (800 cells/well) were seeded into 6-well plates (Corning, USA) for approximately 14 d of culture. The cells were then fixed with 4% paraformaldehyde for approximately 20 min followed by staining with 1% crystal violet for 1 h. All visible colonies (>50 cells) were counted under a microscope.

### 4.9. EdU Assay

Cells (2 × 10^4^ cells/well) were seeded in 96-well plates for 24 h. The cells were then incubated with 50 µM EdU (RiboBio, Guangzhou, China) for 2 h at 37 °C. Paraformaldehyde was applied after washing the cells with PBS. Next, glycine was added, followed by 0.5% Triton X-100 for 10 min after discarding glycine. Apollo^®^ (1×) was used for staining, and nuclei were stained with Hoechst. Finally, pictures were taken under a fluorescence microscope. The cell proliferation ratio was calculated by counting the proportion of cells labelled with EdU.

### 4.10. Wound Healing Assay

T24 and 5637 cells were cultured in 6-well plates. When the cells reached approximately 90% confluence, a standard 200 μL sterilized pipette was used to create a scratch. The cells were then cultured with medium containing 2% FBS. At 0 and 24 h after scratching, pictures were taken in the same position by using an inverted microscope.

### 4.11. Transwell Assay

In the Transwell invasion assay, cells (3 × 10^4^) were seeded into the upper chamber that had been precoated with Matrigel (Corning, USA), and the lower chamber had a higher content of FBS. After 48 h, the cells in the upper chamber were washed with PBS, fixed in paraformaldehyde for 20 min, and stained with 0.1% crystal violet. The inner cells were gently scraped away with a cotton swab. After the chambers were dried, the number of invading cells was counted under a microscope. The experimental methods of the Transwell migration assay were consistent with those of the invasion assay, but the upper chamber was not coated with Matrigel.

### 4.12. Cell Cycle Assay

Cells were centrifuged and then fixed overnight in precooled 75% ethanol. After centrifugation of the sample, the staining solution was added to all samples for 30 min of incubation in the dark. Next, the prepared staining solution was homogeneously mixed with the cells. Finally, flow cytometry was used, and the results were analysed by FlowJo_v10.8.1 software.

### 4.13. Cell Apoptosis Assay

The cells in each sample were washed and then trypsinized without EDTA. Next, the cells were washed with precooled PBS after centrifugation twice. The cells were gently re-suspended with a pre-cooled 1× Binding Buffer. Annexin V-FITC/PI Cell Apoptosis Detection Kit (Servicebio, China) was then used according to the instruction. Finally, the cell apoptosis rate was measured by a flow cytometer.

### 4.14. Reactive Oxygen Species (ROS) Analysis

A ROS Assay Kit (Beyotime, China) was used to detect the levels of ROS. T24 and 5637 cells were cultured in 6-well plates. When the cells grew to 80% confluence, 10 μmol/L DCFH-DA was added to all wells and incubated at 37 °C. for 20 min, and the cells were digested after being washed three times with PBS. Next, the suspension was centrifuged. Finally, the pellet was resuspended in PBS and flow cytometry was carried out for analysis.

### 4.15. Glutathione (GSH) Assay

For GSH determination, a GSH Detection Kit (Jiancheng, China) was purchased, and the level of GSH was measured according to the instructions. First, BCa cells were collected and washed three times with PBS, reagent one was added and cells were lysed by ultrasound. Next, the mixture was centrifuged at 8000× *g* for 10 min, and supernatant was collected and placed at 4 °C. The reagents two and three were added to the above supernatant, and the value at 412 nm was measured with microplate reader. Finally, we calculated the GSH content according to the instructions.

### 4.16. Immunofluorescence (IF) and Immunohistochemistry (IHC)

These procedures were performed as previously described [41].

### 4.17. Xenograft Assay

BALB/c nude mice approximately four weeks of age weighing approximately 20 g were used in this study after purchase from Beijing HFK Bioscience Co., Ltd. (Beijing, China). All mice were randomly divided into two groups: the sh-NC group and the sh-HSPA5 group. All mice were subjected to the next phase of the experiment after acclimation under specific pathogen-free (SPF) conditions for 5 d. Next, 5 × 10^6^ T24 cells in 150 μL of PBS were injected into the forelimb axilla of the mice. All tumors were dissected to measure weight and performed IHC after approximately 35 d.

### 4.18. Statistical Analysis

SPSS Statistics 26, GraphPad Prism 8 and Flow Jo v10.8.1 were used for statistical analyses. A *t*-test was utilized for comparison between two or three groups. The χ^2^ test was used to assess the correlation between protein expression and patient clinical characteristics. The Kaplan–Meier method was used to construct survival curves. The relationships between genes were assessed by using Spearman’s correlation. Unless otherwise stated, all data were the mean ± SD from an average of three experiments. *p* < 0.05 indicates statistical significance.

## 5. Conclusions

In summary, the results of our study demonstrated that HSPA5 was upregulated in BCa and correlated with patient prognosis. HSPA5 knockdown retarded the proliferation, migration and invasion of BCa cells by regulating the VEGFA/VEGFR-2 signaling pathway. In addition, overexpression of VEGFA alleviated the negative effect of HSPA5 downregulation. Moreover, we found that HSPA5 could inhibit the process of ferroptosis through the P53/SLC7A11/GPX4 pathway. Hence, HSPA5 can be used as a novel biomarker and latent therapeutic target in the clinic.

## Figures and Tables

**Figure 1 ijms-24-05144-f001:**
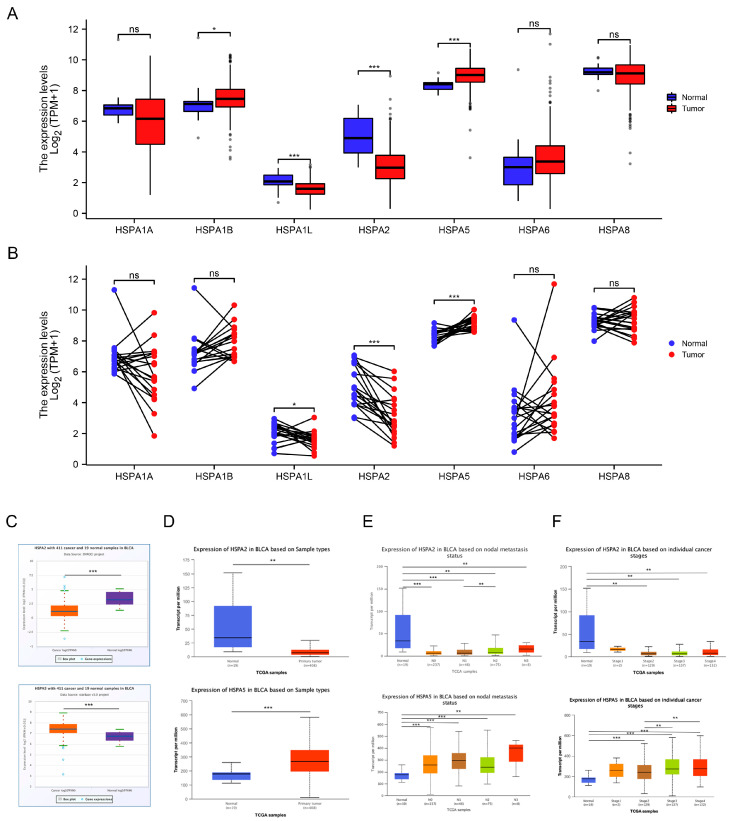
The expression levels of HSP70s in BCa. (**A**,**B**) Expression of HSP70s from unpaired and paired BCa samples from The Cancer Genome Atlas (TCGA) database. (**C**,**D**) Expression of HSPA2 and HSPA5 in BCa from starBase (https://starbase.sysu.edu.cn/) (accessed on 4 January 2022) and UALCAN (http://ualcan.path.uab.edu/analysis.html) (accessed on 4 January 2022). (**E**,**F**) The relationship between protein expression and nodal metastasis status or individual cancer stage. All data are presented as the means ± SD. * *p* < 0.05, ** *p* < 0.01, *** *p* < 0.001.

**Figure 2 ijms-24-05144-f002:**
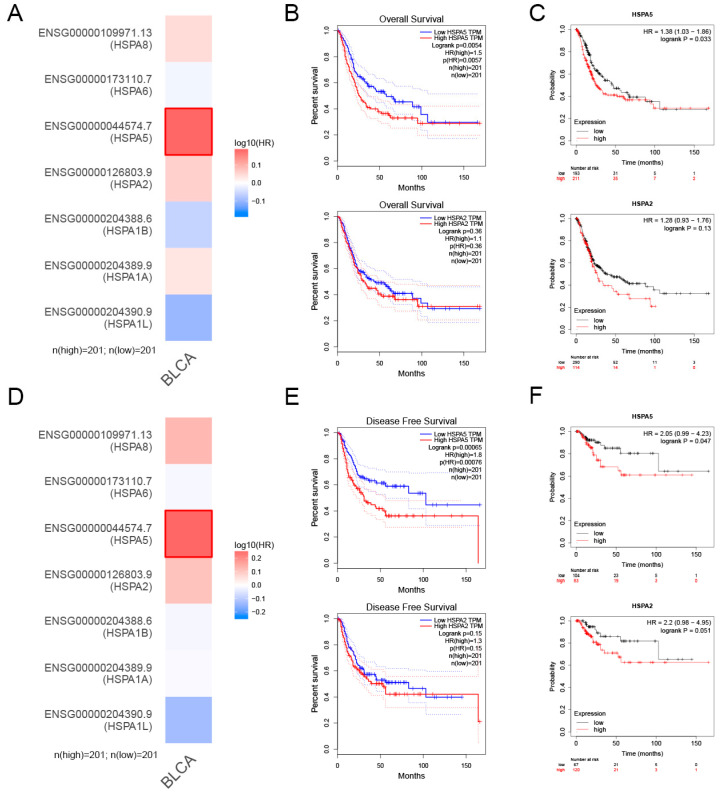
Correlation between HSPA2 or HSPA5 expression and patient survival. (**A**) The relationship between the expression levels of HSPA2 or HSPA5 with patient overall survival from the TCGA heatmap. (**B**,**C**) Analysis of overall survival from GEPIA (http://gepia.cancer-pku.cn/) (accessed on 4 January 2022) and Kaplan-Meier plotter (http://kmplot.com/analysis/index.php?p=background) (accessed on 6 January 2022). (**D**–**F**) Analysis of patient disease-free survival from the TCGA, GEPIA and Kaplan-Meier plotter databases.

**Figure 3 ijms-24-05144-f003:**
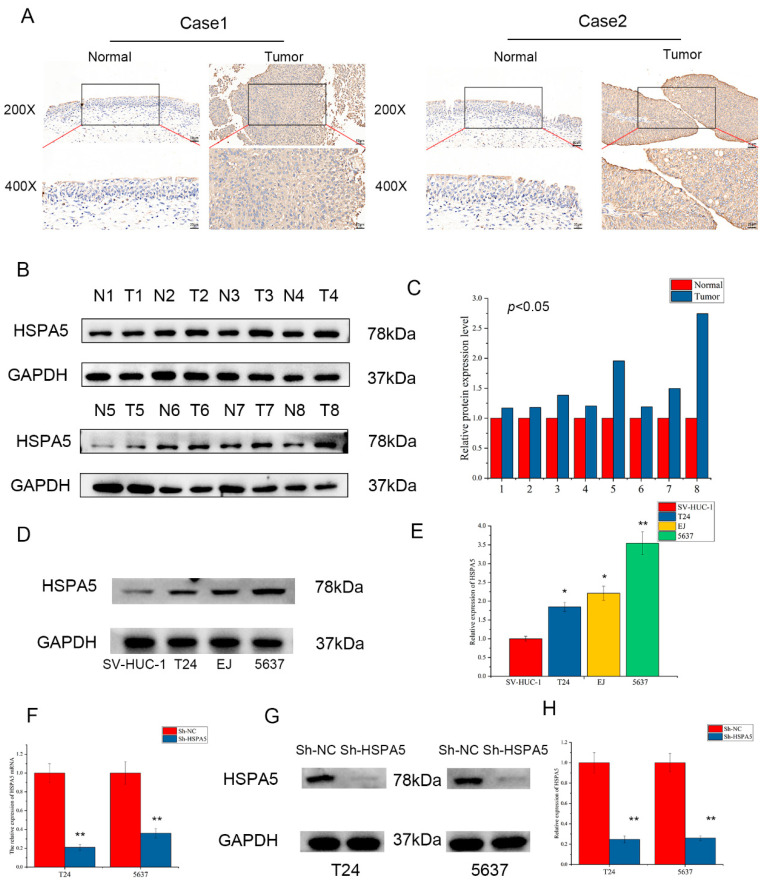
HSPA5 expression in clinical samples and cell lines. (**A**) Immunohistochemical staining observed at 200× and 400× showing the level of HSPA5 in BCa and adjacent tissues. (**B**,**C**) HSPA5 expression in BCa and adjacent samples. (**D**,**E**) The HSPA5 protein level in BCa cell lines and SV-HUC-A cells. (**F**) The expression of HSPA5 mRNA in the sh-HSPA5 groups compared with that in the control groups. (**G**,**H**) The effect of HSPA5 protein knockdown was determined. All data are presented as the mean ± SD. * *p* < 0.05, ** *p* < 0.01.

**Figure 4 ijms-24-05144-f004:**
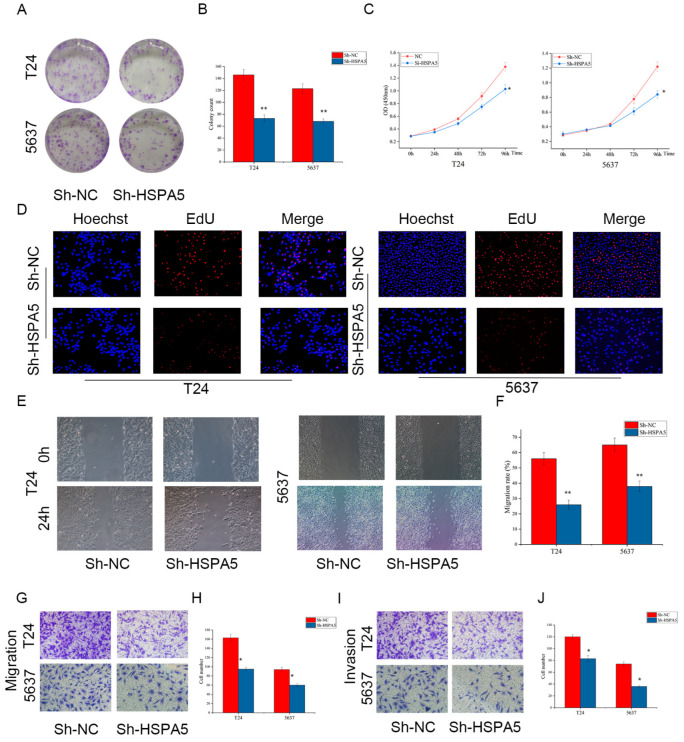
Downregulation of HSPA5 inhibits BCa cell growth and metastasis. (**A**,**B**) HSPA5 downregulation significantly inhibited clonogenic ability. (**C**) Cell viability was measured by CCK-8 assay. (**D**) EdU assay (200×) was utilized to assess cell proliferation. (**E**,**F**) The wound-healing assay (100×) demonstrated that cell migration capacity was inhibited. (**G**,**H**) The results of the migration assay (200×) were consistent with those of the wound-healing assay. (**I**,**J**) Knockdown of HSPA5 decreased cell invasion ability (200×). All data are presented as the mean ± SD. * *p* < 0.05, ** *p* < 0.01.

**Figure 5 ijms-24-05144-f005:**
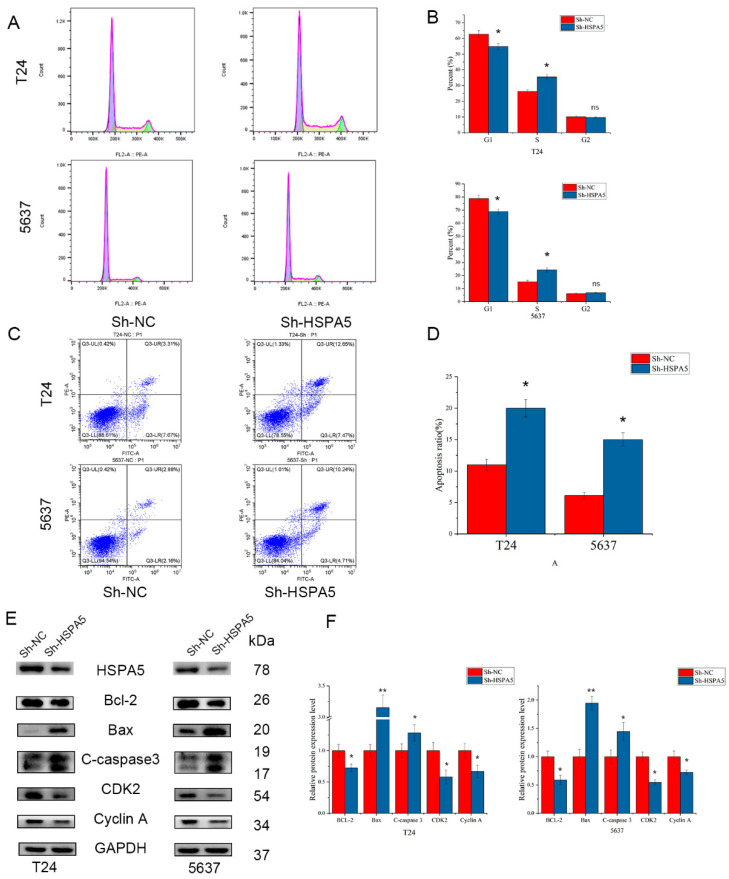
HSPA5 knockdown affects the cell cycle and apoptosis. (**A**,**B**) HSPA5 downregulation increased the percentage of S phase. (**C**,**D**) Low expression of HSPA5 increased cell apoptosis. (**E**,**F**) Western blotting verified the above results. All data are presented as the mean ± SD. * *p* < 0.05, ** *p* < 0.01.

**Figure 6 ijms-24-05144-f006:**
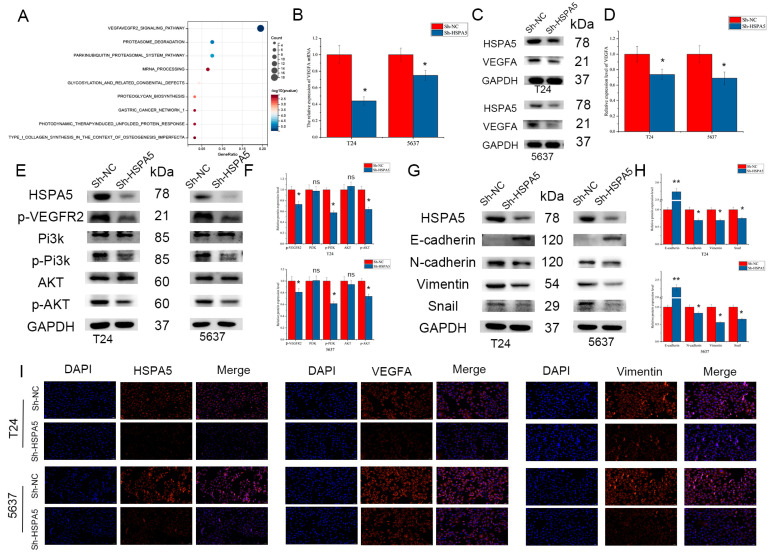
Inhibition of HSPA5 suppresses BCa progression by downregulating the VEGFA/VEGFR2 pathway. (**A**) Signaling pathway enrichment analysis showed that HSPA5 may regulate the VEGFA/VEGFR2 signaling pathway. (**B**) VEGFA mRNA and (**C**,**D**) protein expression after inhibition of HSPA5. (**E**,**F**) HSPA5 could regulate the PI3K/AKT signaling pathway via VEGFA/VEGFR2. (**G**,**H**) HSPA5 is involved in the process of EMT. (**I**) The HSPA5, VEGFA and vimentin expression levels were measured by fluorescence microscopy in the sh-HSPA5 groups and control groups (200×). All data are presented as the mean ± SD. * *p* < 0.05, ** *p* < 0.01.

**Figure 7 ijms-24-05144-f007:**
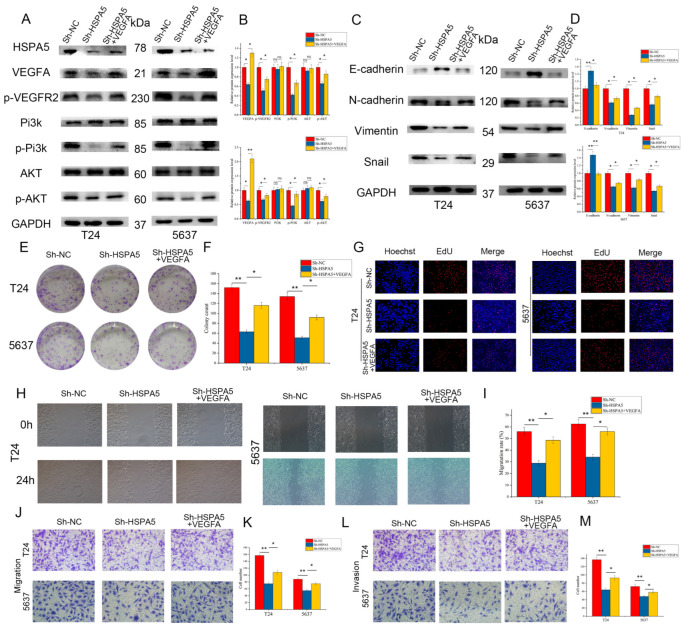
VEGFA reversed the inhibitory effect of HSPA5 knockdown. (**A**,**B**) VEGFA overexpression activated the PI3K/AKT signaling pathway after it was inhibited by HSPA5 downregulation. (**C**,**D**) The expression of EMT-related proteins was measured. (**E**,**F**) Colony formation assay and (**G**) EdU assay (200×) demonstrating that VEGFA promoted BCa cell growth. (**H**–**K**) The migration abilities of the cells in different groups were measured by wound-healing assay (100×) and migration assay (200×). (**L**,**M**) Cell invasion capability increased after transfection with VEGFA (200×). All data are presented as the mean ± SD. * *p* < 0.05, ** *p* < 0.01.

**Figure 8 ijms-24-05144-f008:**
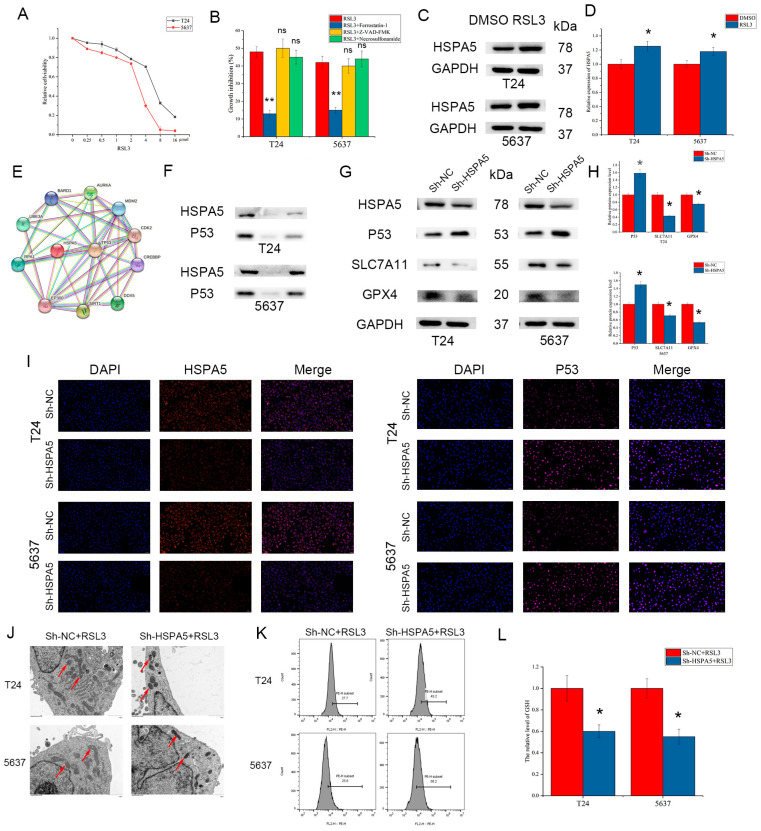
HSPA5 inhibition enhances BCa cell ferroptosis. (**A**) RSL3 (24 h) induced BCa cell ferroptosis in a concentration-dependent manner. (**B**) The inhibitory effects of T24 (5 μmol, 24 h) and 5637 (3 μmol, 24 h) could be decreased by the administration of the ferroptosis inhibitor ferrostatin-1 (2 μmol, 24 h) but not the apoptosis inhibitor Z-VAD-FMK (10 μmol, 24 h) or the necroptosis inhibitor necrosulfonamide (1 μmol, 24 h). (**C**,**D**) The expression of HSPA5 after treatment with RSL3 (1 μmol, 24 h). (**E**) STRING database (https://cn.string-db.org/) (accessed on 27 August 2022) and (**F**) CO-IP analyses revealed that HSPA5 interacted with P53. (**G**–**I**) HSPA5 could regulate the P53/SLC7A11/GPX4 pathway (200×). (**J**) Mitochondrial changes (red arrow) in the cells in different groups. (**K**) ROS and (**L**) GSH levels in BCa cells. All data are presented as the mean ± SD. * *p* < 0.05, ** *p* < 0.01.

**Figure 9 ijms-24-05144-f009:**
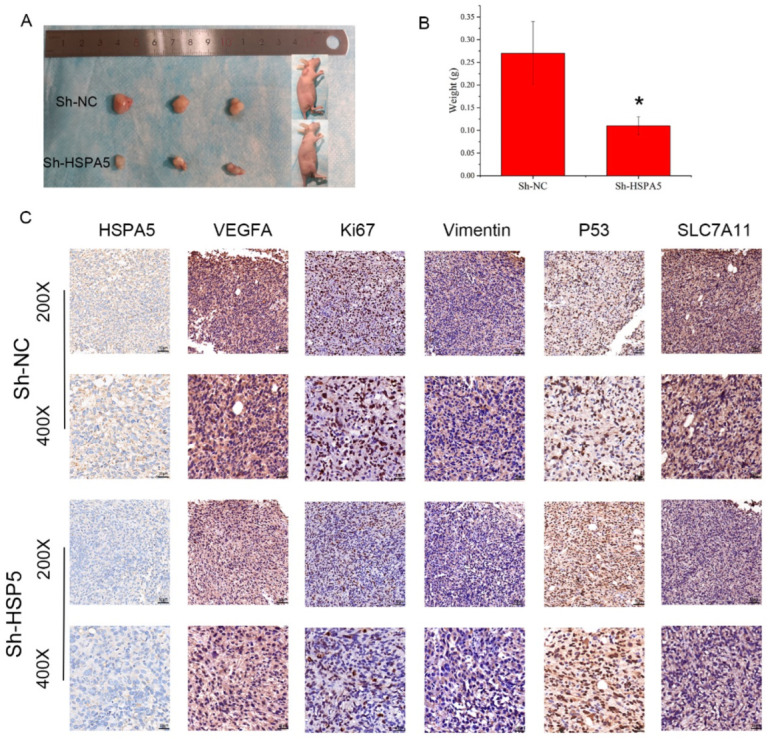
HSPA5 knockdown inhibits xenograft growth in vivo. (**A**) HSPA5 downregulation inhibited the growth rate of xenografts in vivo. (**B**) The tumors weights in the sh-HSPA5 groups were lower than those in the control group. (**C**) IHC results showing the expression of various proteins in tumors (200× scale bar is 50 μm and 400× is 25 μm). All data are presented as the mean ± SD. * *p* < 0.05.

**Figure 10 ijms-24-05144-f010:**
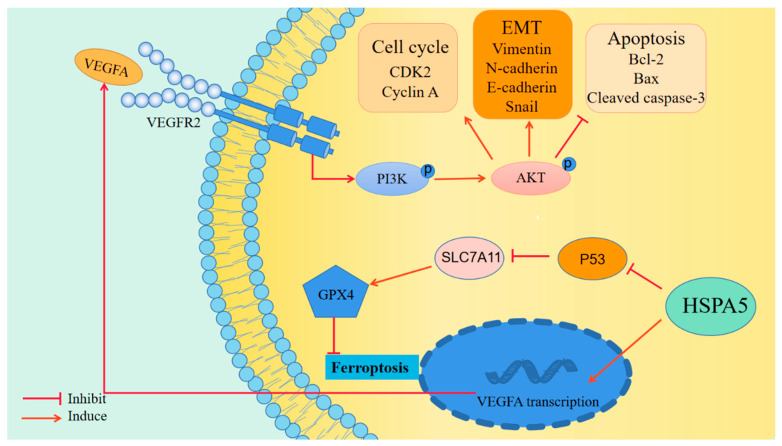
HSPA5 facilitates bladder tumour progression via VEGFA/VEGFR2 pathway and enhances ferroptosis resistance through P53/SLC7A11/GPX4 pathway.

**Table 1 ijms-24-05144-t001:** Primer sequences applied in our study.

	Forward Primer 5′–3′	Reverse Primer 5′–3′
GAPDH	TGACTTCAACAGCGACACCCA	CACCCTGTTGCTGTAGCCAAA
HSPA5	GGGGTGAGGGGAGGGAGTATTTG	GCTGGGAGACTGAGGTGGAAGG
VEGFA	CTTCGCTTACTCTCACCTGCTTCTG	GCTGTCATGGGCTGCTTCTTCC

**Table 2 ijms-24-05144-t002:** Primary antibodies applied in our study.

		Dilution			s
Antibody	Specificity	WB	IHC	IF	Supplier
HSPA5	Rabbit	1:2000	1:100	1:100	Proteintech
VEGFA	Rabbit	1:500	1:100	1:100	Wanlebio
VEGFR2	Rabbit	1:500	-	-	ABclonal
p-VEGFR2	Rabbit	1:500	-	-	ABclonal
PI3K	Mouse	1:1000	-	-	Proteintech
p-PI3K	Rabbit	1:1000	-	-	ABclonal
AKT	Rabbit	1:1000	-	-	ABclonal
p-AKT	Rabbit	1:1000	-	-	ABclonal
E-Cadherin	Rabbit	1:1000	-	-	Proteintech
N-Cadherin	Rabbit	1:1000	-	-	Proteintech
Vimentin	Rabbit	1:2000	1:100	1:100	Proteintech
CDK2	Rabbit	1:1000	1:100	-	Wanlebio
Cyclin A	Rabbit	1:500	-	-	Wanlebio
P53	Rabbit	1:1000	1:100	1:100	Proteintech
SLC7A11	Rabbit	1:1000	1:100	-	Proteintech
GPX4	Mouse	1:1000	-	-	Proteintech
Ki-67	Mouse	-	1:100	-	Proteintech
GAPDH	Rabbit	1:1000	-	-	Servicebio

## Data Availability

Data are contained within the article.

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
