# Peer review of "HSPA5 Promotes the Proliferation, Metastasis and Regulates Ferroptosis of Bladder Cancer"

_ijms, 2023, doi:10.3390/ijms24065144_

Round 1

Reviewer 1 Report (Previous Reviewer 2)

This is now a much improved manuscript in terms of the language and I have a few minor points:

1. Page 3 line 120: It should be % of apoptotic cells and not apoptosis ratio.

2. Your results presented in Fig 5 shows that apoptosis is increased in HSP5 knockdown cells but you agrue that ferropotosis is the main mechanism of cell death. how do you reconcile that? Please provide a valid explanation for this result in your discussion section.

3. Minor remark: Figure 5 legend: A-B It should be HSPA5 downregulation and not depression.

4. EDU images in Fig 4 D are no very clear. Please ensure better clarity in the final images.

5. What is the rationale for using only T24 for in vivo assay? Please provide explanation. 

Author Response

Reviewer 2 Report (New Reviewer)

In this study, the author analyze the expression of HSPA5 expression in  bladder cancer cell line and patient samples. They found HSPA5 functions through VEGFA/VEGFR2 pathway. and they also conclude that HSPA5 expression sensitize the BCa to ferroptosis. This study is well organized and written. However, there are still several questions remaining for the study to be published. Please see the comments below:

(a) The author claimed that in cell lines, knocking down HSPA5 can inhibit VEGFA/VEGFR2 pathway, is this the same mechanism in the tumor sample? Can the author do a Immunostaining? If yes, this will greatly improve the importance of this work.

(b) The author showed that HSPA5 inhibition make cells resistant to ferroptosis. However, there is not enough proof to support the author's point. At least the author should compare the cell death in HSPA5 KD and control cells with RSL3 and Ferrostatin-1 .

(c) For all the images for imunostaining, there should be a ruler.

(d) The quality of all the panels should be improved, which is very important.

(e) For the in vivo studies, there are only 3 mice for each group. This is worried about picking up mice.

(f) For the method part, the author should pay attention to it instead of copy and paste.

Author Response

Reviewer 3 Report (New Reviewer)

The paper is prepared in a way that makes it impossible to verify how the particular steps were performed. There is no information on the number of patients and data on the collected tissues. There is no information about the approval of the bioethics committee. Individual statistical analyzes have not been presented in a way that would enable to check the correct application of selected methods. The description of the methodology is insufficient, for example, in the description of "Quantitative real-time PCR" only the sequencesof the primers are given, whereas there are no conditions for conducting the experiments, information about the equipment, reagents and the method of evaluation.

Round 2

Reviewer 3 Report (New Reviewer)

 carelessly prepared

Author Response

Thank you again for your comments concerning our manuscript. We have carefully considered all comments from you and revised our manuscript accordingly. Please see the attachment.

This manuscript is a resubmission of an earlier submission. The following is a list of the peer review reports and author responses from that submission.

Round 1

Reviewer 1 Report

It is a strange paper. All figures cannot be handled even by extremely enlarged font. All figures are heavy overloaded and cannot analyzed. For example: Survival in patients is given in percentages and months without any number of patients?! The results were written without any data only with reference to figures which cannot analyzed.This paper based on a lot of work but essential data are missed.

Reviewer 2 Report

English is of very poor quality, extensive editing is needed.

Reviewer 3 Report

In this manuscript the authors demonstrated that HSPA5 was upregulated in BCa and correlated with prognostic ability. In the era of precision medicine and targeted therapy these findings are worth mentioning.

Following some comments for major revision.

Please, move lines 62-66 of the Introduction paragraph towards the Discussion section. Usually, the end of the Introduction opens to the main goals of the study or main hypothesis tested by the analysis.

Are the findings or the dysregulation of VEGFA/VEGFR2 signaling pathway correlated with aggressive variant histology bladder cancer?

Conventional surgical-pathological (doi: 10.1007/s00345-021-03776-5; doi: 10.1007/s00345-022-04025-z; doi: 10.1016/j.euros.2022.05.001), immunohistochemical (doi: 10.1016/j.urolonc.2021.10.010; doi: 10.1177/10668969221095173), clinico-laboratoristic (doi: 10.1016/j.urolonc.2021.04.026; doi: 10.1016/j.urolonc.2020.01.015; doi: 10.1159/000477405) are already well-established tools to guide the management of BCa pts. Furthermore, emerging tools including NGS sequencing, ct/cfDNA represent an attractive platform of diagnosis, longitudinal monitoring and could serve as both prognostic and predictive biomarker among bladder cancer patients within different settings (doi: 10.1016/j.euf.2022.04.017; doi: 10.1016/j.ajur.2021.05.001; doi: 10.1016/j.euo.2020.01.003; doi: 10.1097/MOU.0000000000001013). The authors should consider to discuss and compare their findings with these ready-to-use tools in the clinical daily practice to improve the scientific and clinical sound of the manuscript.

The authors should consider an english review by a native speaker.